# An Efficient Attribute-Based Access Control (ABAC) Policy Retrieval Method Based on Attribute and Value Levels in Multimedia Networks

**DOI:** 10.3390/s20061741

**Published:** 2020-03-20

**Authors:** Meiping Liu, Cheng Yang, Hao Li, Yana Zhang

**Affiliations:** 1State Key Laboratory of Media Convergence and Communication, Communication University of China, Beijing 100024, China; liumeiping@cuc.edu.cn (M.L.); zynjenny@cuc.edu.cn (Y.Z.); 2School of Information and Communication Engineering, Communication University of China, Beijing 100024, China; 3Post-Doctoral Research Center of Zhuhai Da Hengqin Science and Technology Development Co. Ltd., Zhuhai 519000, China; cuclihao@cuc.edu.cn

**Keywords:** policy evaluation, ABAC, binary identifier, depth index, policy retrieval

## Abstract

Internet of Multimedia Things (IoMT) brings convenient and intelligent services while also bringing huge challenges to multimedia data security and privacy. Access control is used to protect the confidentiality and integrity of restricted resources. Attribute-Based Access Control (ABAC) implements fine-grained control of resources in an open heterogeneous IoMT environment. However, due to numerous users and policies in ABAC, access control policy evaluation is inefficient, which affects the quality of multimedia application services in the Internet of Things (IoT). This paper proposed an efficient policy retrieval method to improve the performance of access control policy evaluation in multimedia networks. First, retrieve policies that satisfy the request at the attribute level by computing based on the binary identifier. Then, at the attribute value level, the depth index was introduced to reconstruct the policy decision tree, thereby improving policy retrieval efficiency. This study carried out simulation experiments in terms of the different number of policies and different policy complexity situation. The results showed that the proposed method was three to five times more efficient in access control policy evaluation and had stronger scalability.

## 1. Introduction

Today, Internet of Things (IoT) and multimedia have many cross-integrations. IoT seeks intelligence and can provide technology and platform support for multimedia. Multimedia can make the intelligence of IoT more intuitive and friendly. Therefore, Internet of Multimedia Things (IoMT) technology came into being, making users feel the charming experience brought by intelligent multimedia. In particular, the differences in network composition, transmission media, terminal equipment, applications, and protocols have produced heterogeneous multimedia networks. IoMT, as a novel technology in which smart heterogeneous multimedia things can interact and cooperate with one another and with other things connected to the Internet, facilitates the convenient application of multimedia-based services and applications [1,2]. For example, cloud TV not only provides intelligent video services and human–computer interaction functions, but can also realize the interconnection of multiple information terminals as well as the centralized display and multimedia control of other household appliances.

The characteristics of IoT technology [3] support the sharing and convenient access of multimedia services [4], but because of the openness, heterogeneity, and multiple security domains of the IoMT environment, issues of security, privacy, and trust remain a challenge. Multimedia in IoT services should provide robust and resilient security platforms and solutions against any unauthorized access [5,6]. In many security threats and vulnerability attacks, most of them originate from the unauthorized access of illegal users and ultra vires access of resources by legitimate users. For example, the malicious tampering of WhatsApp and Telegram multimedia data was due to the privilege settings of external storage and internal storage, which led to the hackers having internal privileges where they can access the user’s personal information such as dialogue, photos, videos, and contact list [7]. Access control technology aims to solve this problem. In access control, access requesters’ rights are regulated and restricted through the identity information of visitors and the security policy provided within the system. This ensures that resources and services can be accessed and used by legitimate users in a complex network environment, while preventing illegal users from stealing and abusing resources and services, and unauthorized access by legitimate users [8].

Due to the openness of the multimedia networks and the intelligence of IoT, users can perform real-time human–machine interaction anytime and anywhere to enjoy intelligent services, which undoubtedly increases risks. It is necessary to analyze the security access requirements of a multimedia network in terms of its roles, environments, resources, and many other aspects to control the user’s authority by multiple-factor restrictions (i.e. more fine-grained management and control). Traditional access control models such as Discretionary Access Control (DAC) [9], Mandatory Access Control (MAC) [10,11], and Role-Based Access Control (RBAC) [12] cannot meet these actual needs. At the same time, the dynamics of a multimedia network make the current user needs and computer environment change more frequently. It is urgent to improve the robustness of the access control model and the dynamics of authorization. However, the static authorization method of the traditional access control model affects the scalability and adaptive change of permission grants. For the coexistence of multiple security domains in IoMT, the traditional access control model is limited to the relatively closed network environment, so that access control policies cannot be applied to multiple security domains.

Attribute-Based Access Control (ABAC) [13,14] abandons the traditional single-factor constraints and the inflexibility of authority control, and can solve the fine-grained problems faced by resource protection in a multimedia network as well as the flexibility and dynamics of authority control. It also has the support of the extensible access control markup language (XACML) framework, which provides an ideal access control scheme for IoMT. With the increase in multimedia services provided by the IoT platform, the scale of users and data in multimedia networks is increasing. The corresponding access control system is becoming more and more complex. The number of access control policies is so large that the authorization needs to traverse all policies, which is inefficient and affects the service provision. However, intelligent multimedia applications need shorter response times to improve performance to provide high-quality services [15,16]. It is urgent to improve the access control policy evaluation efficiency to improve the performance of the access control system. In this regard, researchers have made a series of improvements [17,18] and applied the optimization scheme to specific scenarios such as industrial network systems [19] and medical information systems [20,21,22], but without an intelligent multimedia network. Meanwhile, most schemes to improve the efficiency of access control policy evaluation ignore the research of policy retrieval.

In order to solve the above problems, this paper proposed a policy retrieval method based on ABAC to improve the efficiency of access control policy evaluation in a multimedia network to improve the overall performance of the system. The rest of this paper is organized as follows. Section 2 summarizes the access control scheme in existing multimedia networks and the research progress of ABAC policy evaluation; Section 3 introduces the policy retrieval method based on a binary identifier at the attribute level and the policy decision tree retrieval method based on the depth index in the attribute value level; Section 4 presents the experimental results of the above algorithm and compares it with other algorithms. Finally, our conclusions are presented in Section 5.

## 2. Related Work

As an important part of information security technology, access control technology has been widely used in multimedia networks. Parwinder et al. [23] proposed a secure multi-factor remote user authentication scheme in a multimedia environment, which could resist all active and passive attacks, but lacked a certain permission granting mechanism. Kai Fan et al. [24] proposed an access control scheme based on multi-authority attribute encryption for media big data, focusing on cross-domain access control. The improvement of system efficiency is due to the use of trusted agents to reduce the computational overhead of ciphertext re-encryption, rather than to force it in policy retrieval. Changsha Ma et al. [25] implemented access control for media sharing based on ABE, emphasizing the effectiveness of key management and access security. However, the improvement of efficiency was for encryption and decryption rather than policy evaluation. Liang et al. [26] designed a general media-aware security architecture for the security heterogeneity of multimedia applications in the Internet of Things. However, the above schemes lack personalized permission control for users in multimedia networks and do not consider the efficiency of access control. In this paper, the new ABAC model was applied to the multimedia networks to ensure data security. However, the low efficiency of ABAC policy evaluation is always the key factor to restrict the system performance.

Aiming at the inefficiency of ABAC policy evaluation, researchers have also made some other achievements. Alex et al. [27,28] proposed XEngine, which numeralizes text policies and converts them into numerical policies with normalized structure to handle requests more effectively, but its numeralization cannot completely replace the complexity of XACML policy description. Santiago P et al. [29] constructed a new data structure based on the XEngine tree, and provided an evaluation method based on a binary search. The Enterprise XACML Policy Evaluation System [30] has a built index structure according to attribute tags in the policy target. It reduced policy retrieval space and improved policy evaluation performance, but could lead to the policy repeating the index due to duplication of the attribute tags. Feng [31] reduced policy retrieval space through the attribute list and adopted the XML database to accelerate the policy analysis, but sacrificed the fine-grained control of the policy. The team then proposed rule refinement as a policy optimization technology to assist other engines to improve the efficiency of policy evaluation. Niu et al. [32] adopted the multi-level caching mechanism based on statistical analysis, and reordered the policies based on statistical analysis, which reduced the calculation of policy matching and improved the overall evaluation performance. Qi [33] proposed an adaptive policy reordering method, which ranked the high-priority policies in front of them, thereby improving the efficiency of policy evaluation. Due to the reordering of policies, the above two methods are only applicable to the combination algorithm of Permit-overrides and Deny-overrides, but not to First-applicable.

Most of the references above-mentioned ignore the policy retrieval in the process of policy evaluation, so the efficiency improvement is not high. Some references have used the policy retrieval method, but the fine-grained permission control and the complete coverage of the policy combination algorithm have not been considered. In light of these shortcomings, considering the applicability of the algorithm and the architecture of the multimedia networks, an ABAC policy retrieval method based on attribute and value levels was proposed on the basis of the XACML framework [34]. At the attribute level, this paper used the logical and calculation based on binary identifiers to retrieve the policies satisfying the access request and generate a new policy set. At the attribute value level, the policy decision tree retrieval method based on the depth index was adopted. This method can filter the policies with mismatched attribute values, reduce the retrieval scope, and speed up the matching speed of policies with access request, thus improving the efficiency of policy evaluation, ensuring the efficiency of users using IoMT services and resources.

## 3. Improved Evaluation Algorithm

### 3.1. Related Theory of Attribute-Based Access Control (ABAC)

#### 3.1.1. Basic Concepts

ABAC obtains three kinds of entity attribute information: subject, object, and environment. Through the description and calculation of attribute expression, it triggers the corresponding access control policies and evaluates them to realize the management of resource access. Policy evaluation is the process that the authorization engine matches the access control policies when it judges whether to grant certain operation permissions to the corresponding subject. The evaluation results can be “permit”, “deny”, “indeterminate”, or “not applicable”. In this paper, the basic concepts are formally described referring to the existing related literatures [19,22,35,36].

**Definition** **1**(Entity)**.**
*The object is embodied as two parts of resource and operation. Then, there are four entities, and a tetrad (S, R, O, E) that ABAC can be abstracted into, where S, R, O, and E represent the entity set of subject attributes, resource attributes, operation attributes, and environment attributes, respectively. These four entity sets can be expressed as S = {s1,s2,s3…,sm}, R = {r1,r2,r3…,rn},O = {o1,o2,o3,…,oj},E = {e1,e2,e3…,ek},where n,m,k,j ≧ 1.*

**Definition** **2**(Attributes)**.**
*These are used to describe the intrinsic characteristics of entities. The set of attributes can be represented as A = {a1, a2, a3…, as}. SA, RA, OA, and EA respectively represent the set of all subject attributes, resource attributes, operation attributes, and environment attributes, denoted as SA = {sA1,sA2,sA3,...,sAm}, RA = {rA1,rA2,rA3,...,rAn}, OA = {oA1, oA2, oA3,...,oAj}, EA = {eA1,eA2,eA3,...,EAk}. Take SA as an example, sAi represents the set of the attributes for subject si, where sAi⊆A.*

**Definition** **3**(Attribute-value pair)**.**
*The description of attribute and its specific value is represented by attribute-value pair (avp), which is defined as a two-tupies (attribute, value) (i.e. attribute = value). Savp, Ravp, Oavp, and Eavp are used to represent the attribute-value pairs of subject, resource, operation, and environment attributes, respectively.*

**Definition** **4**(Policy)**.**
*This is denoted by p and describes a subject with specific attribute values, which allows or deny the operations on the resources of the corresponding attribute values in a specific environment. Its formal description can be expressed as sign← (SA, RA, OA, EA), and the value of sign is permit or deny, indicating positive authorization and negative authorization, respectively. The policy set is expressed as P = {p1, p2, p3, ..., pn}.*

**Definition** **5**(Rule)**.**
*This is denoted by r and is the basic unit of the policy and the smallest unit to execute the policy evaluation. The value of its effect element is permit or deny, which represents the authorization result of the rule.*

#### 3.1.2. Policy Decision Based on XACML

XACML defines the standard representation format of the policy and the standard method of making authorization decisions, making the access control policy evaluation based on XACML universal and widely used in ABAC research. XACML provides many data types, matching functions and combinational logic for access control policy evaluation, which has rich and intuitive semantics. Currently, common XACML standards in practical applications include XACML3.0, Sun XACML, and so on. Based on the Sun XACML framework, this paper develops the algorithm, and uses XACML3.0 standard to write the access request and strategy in the multimedia network environment.

The data flow model of the XACML framework is shown in Figure 1, which contains components Policy Decision Point (PDP), Policy Enforcement Point (PEP), Policy Administration Point (PAP), and Policy Information Point (PIP). PEP receives the user request, then the Context handler converts the original request into XACML format, and sends the XACML request to PDP. When PDP processes the XACML request, it needs to obtain the policy or policy set from PAP. Meanwhile PDP obtains the entity attributes required by the request from PIP through the Context handler and then it evaluates the policy to get the results and feeds back to the Context handler. The Context handler converts the XACML response format to the PEP format and returns it to the PEP, which performs the corresponding decision result and allows access to the resource if it is positive authorization [22].

### 3.2. Policy Retrieval Based on Binary Identifier

In the era of IoT, people need fast, efficient, low-power, anonymous, and intuitive technology to obtain services and data, so that life is more digital and intelligent. In IoT video cloud platform, through the IoT protocol, one or more video streaming media services are registered and connected to the same platform to provide users with cross network limited video streaming media services. ABAC based on XACML guarantees the confidentiality and integrity of video stream and realizes the security management of media resources. At the same time, the fine-grained control of ABAC on resources leads to the increase of access control policies and attributes, which has become a key factor restricting the efficiency of policy evaluation, thereby affecting the real-time and efficiency of video on demand. According to the related theory of ABAC and XACML 3.0 standard, when the user requests the video streaming media services, all related attributes are invoked to be parsed and matched to all policies. This means that XACML based access control policy evaluation needs to traverse all policies for judgment, in which attribute matching not only considers matching function, but also the uniform resource name (URN) and data type of attribute. All of the above matching methods are string matching, and only if all the attribute information matches, will the corresponding authorities be given, so it is inefficient. In addition, if a policy contains more attributes, in the process of matching attributes one by one, it will not be found inconsistent with the request until the last attribute is matched. If there are many such policies, it will greatly increase the evaluation time and reduce the efficiency.

To solve the above problems, this section proposes an attribute-level policy retrieval method based on binary identifiers. Count all the attributes involved in the whole access control system and arrange them in a certain order. Mark the attribute in the access request or policy with 0 or 1, respectively, with 0 indicating that the attribute is not included, and 1 indicating inclusion, thus forming a binary identifier marking the attribute set of the access request or policy. Through the logical and operation between the binary identifier of the access request and the identifiers of all policies, an efficient policy retrieval is realized, which avoids redundant string matching. If the result is consistent with the policy identifier, it indicates that the policy may meet the access request, that is, the attributes required by the policy are provided in the user request.

In the IoT video cloud platform, when users request limited streaming media services through cloud TV, the access control system will collect attribute information of corresponding users and resources, some attributes and the attribute value set of each attribute is shown in Table 1. The policies and access requests are marked according to the attribute order in Table 1. Table 2 shows some of the policy descriptions and their binary identifier representations in the access control of the video cloud platform. If the set of user request attributes is {SA_ role = common member, RA_ type = no titles and tail, RA_ quality = SD, OA_ type = watch}, its binary identifier is 10010110. By the logical and operation between 10010110 and the identifiers in Table 2, it can be seen that the fourth-row policy of Table 2 is the policy that may satisfy the request.

In order to implement the above algorithm, the XACML components need to be extended. When presetting the access control policies in the experiment, the binary identifier and the sequence number are used as the policy unique identifier PolicyID. We designed a logic operation module in PDP; when a user requests access to video resources, PDP generates the binary identifier of the request and passes it to the logical operation module after collecting the relevant attribute information of the user from PIP. At the same time, the logical operation module extracts the binary identifier from the PolicyID of each policy from PAP. Then, it performs logic and operation between these identifiers and request binary identifier to retrieve the policies satisfying the request. If the result is consistent with the policy identifier, the corresponding policies in PDP are cached, and a new policy set is generated. Then, the PDP only needs to evaluate the new policy set whose scale is reduced and passes the evaluation results to the PEP.

### 3.3. Policy Decision Tree Retrieval Based on the Depth Index

The policy retrieval based on the binary identifier found the policies that fully met the access request at the attribute level, thus reducing the scale of the policy, but the new policy set still has many inconsistencies at the attribute value level. In particular, after analyzing the attribute values of each attribute one by one, we found that the last one was inconsistent with the request, which wastes a lot of time. If there are many such invalid policies, the retrieval time of the policy will increase, which will lead to the decrease in evaluation efficiency. This section presents the policy decision tree retrieval method based on depth index, which abstracts policies to a decision tree and reconstructs the decision tree with the depth index of each attribute. Using the reconstructed policy decision tree for retrieval can save time and quickly match the policies that meet the request, thereby improving the efficiency of policy evaluation.

#### 3.3.1. Construction of Policy Decision Tree

In data structure, a decision tree is an abstract concept used to prove lower bounds, usually a binary tree, where the root node represents all possible permutations and each node represents a set of possible permutations between elements. Each of the algorithms that sorted by using only comparison can be represented by a decision tree, so the policy retrieval based on attribute values in this section meets this situation. Therefore, all policies in the policy set can be abstracted into a decision tree by structuring the ABAC entity attributes. The policy set here is the new policy set generated after policy retrieval based on binary identifiers, that is, all policies (or all policies of the subset) have the same attributes.

**Definition** **6**(Policy decision tree)**.**
*Suppose there are m attributes in total, then the policy decision tree is a complete tree with a depth of m, the root node represents the policy set, and each node with different depths represents the attribute values of different attributes. Each path from the root node to the leaf node is a possible policy, and such a decision tree is called a policy decision tree.*

To facilitate labeling in the policy decision tree, an example of the policy decision tree is given by replacing the attributes and attribute values in the video services above with letters. As shown in Figure 2, the set of attribute values for each attribute is: A = {a1, a2}, B = {b1, b2, b3}, and C = {c1, c2, c3, c4}. The full permutation of these attributes and their attribute values constitutes |A||B||C| = 24 policies (i.e., there are 24 paths from root node to leaf node in this policy decision tree). In fact, some permutations may not conform to the actual situation, so the number of policies is less than the full permutation. When PDP performs the evaluation, it will retrieve from the root node and filter the paths with mismatched attribute values depth by depth, according to the attribute value in the access request until the appropriate policies are found. By filtering some invalid policies, the algorithm omits the analysis of the attribute values in the remaining depths of these invalid policies, realizing fast retrieval and improving evaluation efficiency.

#### 3.3.2. Policy Decision Tree Based on Depth Index

Using a policy decision tree to realize fast retrieval is mainly to filter some policies according to the attribute value mismatch of some attributes, so the order of using attributes is a problem. This section introduces the concept of the depth index to determine the order of attributes used in policy retrieval. Assuming that the number of policies is n and an attribute has m attribute values, n/m means the average number of matches for this attribute. The smaller n/m is, the fewer matching policies and the more policies are filtered. When n is certain, m must be larger to filter more policies, which means that there are more types of attribute values for this attribute.

**Definition** **7**(Depth index)**.**
*Different depths of the policy decision tree represent different attributes, and the depth index is the size of the set of attribute values on this depth (i.e., the number of attribute values). The depth index is used to represent the depth of attributes in the policy decision tree. The larger the value, the lower the depth and the higher the priority used.*

The structure of the policy decision tree shown in Figure 2 may be varied, and the order of policy retrieval can be A–B–C, A–C–B, B–A–C, etc. After introducing the depth index, the structure of the policy decision tree is shown in Figure 3.

#### 3.3.3. Analysis of the Policy Decision Tree Retrieval Method Based on Depth Index

Taking the policy and attribute information described in Figure 2 as an example, suppose that the attribute information of the user request as {A = a2, B = b3, C = c4}. According to the retrieval method of the policy decision tree, the policy will be filtered out when the attribute values do not match. However, the number of policies that need to be retrieved varies with different orders of using attributes when filtering policies.

It can be seen from Table 3 that the total retrieval number of C–B–A was the least. This is because the ratio of the attribute value on each depth matching with the user attribute is the smallest in this scheme so more mismatched policies can be filtered during retrieval. Additionally, the policy retrieval at the next depth will be affected by the policy filtering at the previous depth. With the minimum number of retrievals at each depth, the total retrieval number will be the minimum, which is the best retrieval scheme.

Therefore, in order to improve the evaluation efficiency in the attribute value level, the number of policy retrieval should be reduced. It is necessary to determine the order of attributes used in policy retrieval, which requires the depth index. Among the three attribute values of a2, b3, and c4 in the above example, the number of policies matching c4 is the least and the depth index of the attribute C, to which c4 belongs, is the largest, followed by B and A, so C–B–A is the best scheme. We used mathematical induction to demonstrate the effectiveness of the scheme in the case of more attributes and a larger set of attribute values.

*Proof:* Suppose there are *n* policies and *m* attributes, and the depth index of each attribute is *d1*, *d2, d3, …, dm,* respectively. In this order, first, it is necessary to retrieve the first attribute value of all policies, so the retrieval times is *n*. Assume that only one policy’s attribute value does not match the request in each retrieval, at least (*d*_1_ − 1) policies can be filtered out after the first retrieval. By analogy, the total retrieval number is:(1)$$\begin{array}{l} mn-\left(m-1\right)d_{1}-\left(m-2\right)d_{2}-\ldots -2d_{m-2}-d_{m-1}+\left(1+2+\ldots +\left(m-1\right)\right)\\ =mn+\frac{m\left(m-1\right)}{2}-\sum_{i=1}^{m-1}\left(m-i\right)d_{i} \end{array}$$

When *m* and *n* are certain, the total retrieval number depends on the formula $\sum_{i=1}^{m-1}\left(m-i\right)d_{i}$. To minimize the retrieval times, the sum of this formula must be the largest, so each addend $\left(m-i\right)d_{i}$ must be the largest. Meanwhile, each factor of the product is required to be the largest, that is, the attribute with a larger depth index *d_i_* has a smaller depth *i*. It can be seen that the policy decision tree retrieval based on depth index can filter more invalid policies and achieve fast matching, thus improving the efficiency of policy evaluation.

In summary, the policy retrieval based on binary identifiers reduced the scale of the policy, and the policy decision tree retrieval based on the depth index achieved the rapid matching of the policies. Therefore, the policy retrieval method based on attribute and attribute value levels proposed in this paper had a higher efficiency, and the two retrieval methods improved the efficiency of policy evaluation at the levels of attribute and attribute value, respectively. The implementation process is shown in Figure 4.

## 4. Experimental Results and Analysis

Based on the format of the XACML official conformance test dataset and XACML3.0 standard, this section constructs some policies and requests by using the attribute and attribute value sets in Table 1 to carry out a series of simulations. Among them, according to the actual needs of users in the IoT video cloud platform, 350 policies were written, each of which contains at least one rule and the rules totaled 500. Different access requests were written according to different resources and environments. In the policy complexity experiment, after adjusting the avp of the above policies, 50 policies were written for each complexity. The experiment showed the availability and effectiveness of the proposed policy retrieval method. Compared with other existing algorithms, it had a higher efficiency and scalability. The experimental environment was as follows: Intel(R) Core (TM) 2 Quad CPU Q9550 @2.83GHz; (RAM) 4.00 GB; Windows7 Ultimate, SP1; Java jre1.8.0.

### 4.1. Analysis of Experimental Results

Statistics were made on the time consumption of each algorithm when the number of rules in the policy increased and the complexity of the policy increased during the simulation. All data were the average results of multiple experiments on multiple requests. The policy complexity refers to the number of avp included in the policy; the policy analysis time refers to the time consumed by the PDP to analyze the attributes and their URN, attribute values, and their data types step-by-step in the policy. The policy evaluation time refers to the total time that PDP retrieves the policies to find the policies matching the access request and gives the response result.

In the following experimental results, Sun XACML refers to the classic algorithm that needs to traverse all the policies in the policy set when evaluating the policy. B-Sun XACML adopts the policy retrieval method based on the binary identifier. B-D-Sun XACML adopts the policy retrieval method based on attribute and value levels, which superimpose the policy decision tree retrieval method based on depth index on B-Sun XACML. The experimental comparison was based on the policy evaluation for the same access request. The more the number of policy retrievals, the longer the time spent.

The results above show the time consumption of policy analysis and evaluation when the number of rules increase. These series of experiments increased the number of rules contained in the policy from 100 to 500, and averaged the results of five experiments on multiple requests. From Figure 5a and Figure 6, it can be seen that the efficiency of the algorithm in this paper has obvious advantages, where the analysis efficiency was 2–3 times higher than the classical algorithm and for evaluation efficiency, B-Sun XACML was twice as efficient, while B-D-Sun XACML was 3 to 5 times higher. Figure 5b shows the difference of policy analysis time between B-D-Sun XACML and B-Sun XACML. The time consumption of the classical algorithm increased with the increase in the number of rules, while the algorithm in this paper tended to be flat, so the scalability was relatively strong.

Figure 7 and Figure 8 show the time consumption of policy analysis and policy evaluation of each algorithm when the complexity of the policy increases. The experiment adopted eight attributes described in Table 1; if the complexity of each policy is eight, the policy retrieval based on the attribute level is meaningless, so the maximum is seven. There were 50 policies for each complexity in the experiment, so the complexity ranged from 150 to 350. It can be seen from Figure 7a that the analysis efficiency of the algorithm in this paper was still twice as high as that of the classical algorithm at the maximum complexity, so it can be seen that the algorithm in this paper also had great advantages in dealing with the complexity of the policy. Figure 7b clearly shows the small difference in analysis efficiency between B-D-Sun XACML and B-Sun XACML. Figure 8 shows that B-Sun XACML was twice as efficient as Sun XACML, while B-D-Sun XACM was 2 to 3 times higher. It can be seen that the double-level policy retrieval algorithm in this paper had more obvious advantages in evaluating efficiency, and due to the slow growth evaluation time, the algorithm had strong scalability.

The policy analysis time of the classical algorithm in Figure 7a decreased with the increase in the complexity of the policy. This is because the policy complexity is mainly represented by the number of avp in the policy. When the complexity is low, the proportion of avp matching between the access request and the policy is small, so more invalid rules are parsed, which makes the time consumption large. When the complexity increases, the avp matching proportion increases, the invalid rules decrease, and the time consumption decreases. However, the algorithm in this paper preferentially filters at the attribute level, avoiding the situation that the attributes do not match, so the analysis time tends to be stable. The analysis and proof are given below.

Table 4 summarizes the proportion of the avp mismatch between five requests and policies with different complexity in the policy complexity experiment. It can be seen that as the complexity increases, the proportion of avp mismatch becomes smaller.

Figure 9 shows the experimental verification for the above analysis. Count the policy analysis time of Sun XACML, in the case of request and policy attributes, exactly matched and incompletely matched, respectively. It can be seen that in the case of attribute perfect match, the analysis time increased slowly with the increase in complexity. For the incomplete match, the proportion of avp mismatch decreased with the increase in complexity, and the analysis time showed a decreasing trend. However, the time consumption at the maximum complexity was still greater than that in the case of attribute perfect match, so the priority retrieval in the attribute level proposed in this paper is feasible and effective.

### 4.2. Time Complexity Analysis

In this paper, the efficiency of evaluation was improved in the part of policy retrieval. Therefore, the evaluation time of policy was divided into retrieval and analysis time, and the time complexity of the existing algorithm was statistically analyzed to quantitatively analyze the efficiency of the policy evaluation. Suppose there are *n* policies in total, and there are *n1* binary matches, the number of attributes is *m*, the retrieval time of each policy is *t1*, the analysis time is *t2*, the binary matching time is *t3*, and the time of each attribute analysis by the policy decision tree is t’. Where *t1*, *t2*, and *t’* are in milliseconds, with an order of magnitude of 1; *t3* is an order of magnitude of 1, which is in microseconds. Algorithms in [27,28,29,30] had a shorter time of policy analysis to improve policy evaluation efficiency, and their analysis time *t2’* was the same order of magnitude as *t2*, but *t2’ < t2*.

Sun XACML : *T1* = *(t1 + t2)n*Enterprise XACML/XEngine : *T2*
*=*
*(t1 + t2’)n*B-SunXACML: *T3*
*=*
*t3n + n1 (t1 + t2)* (in policy level)        *=*
*t3n + n1mt’ + n1t2* (in attribute level)B-D-Sun XACML: *T4*
*=*
*t3n + (n1+n2+n3+…+nm)t’ + nbt2*

For the above algorithm, the time complexity is compared in terms of policy, and it is easy to see that *T2 < T1*. When the order of magnitude of n is greater than 1 and gradually increases, due to the difference in magnitude as well as the number of binary matching policies *n1 < n*, there will inevitably be *T3 < T2*. B-D-Sun XACML adopts the policy decision tree structure, which performs policy retrieval through attribute values. Here, the comparison with B-Sun XACML is in terms of attribute, where *n1, n2, ..., nm* refers to the number of remaining policies after each retrieval and filters out the invalid policy, which is inevitably smaller, and nb represents the number of policies that are finally matched. Since *(n1 + n2 + n3 +... + nm) < n1m, nb < nm << n1*, there is certainly *T4 << T3*, and it can be seen that B-D-Sun XACML had the lowest time complexity and highest efficiency.

## 5. Conclusions

Access control is an important information security technology, which can guarantee the confidentiality and integrity of data. It provides a solution to prevent any unauthorized access for multimedia applications in IoT services. However, the policy evaluation efficiency of access control in the multimedia network is the main factor that restricts the performance of the IoMT device and application. In order to improve the efficiency of policy evaluation, this paper proposed a policy retrieval method based on binary identifiers at the attribute level, and found the policies that may satisfy the access request to generate a new policy set. On this basis, a policy decision tree retrieval method based on depth index at the attribute value level was proposed to continuously reduce the scale of policy. It can improve the efficiency of retrieval and achieve rapid matching to improve the policy evaluation efficiency of IoMT and the performance of device application. Experiments verified the availability and effectiveness of the proposed algorithm. Meanwhile, the evaluation efficiency of the algorithm in this paper is more efficient and scalable.

Subsequently, security in IoMT will be studied on the basis of guaranteeing the efficiency of policy evaluation. It will consider combining the algorithm with attribute-based encryption (ABE) to better guarantee the security of legitimate users’ access to resources.

## 6. Patents

## Figures and Tables

**Figure 1 sensors-20-01741-f001:**
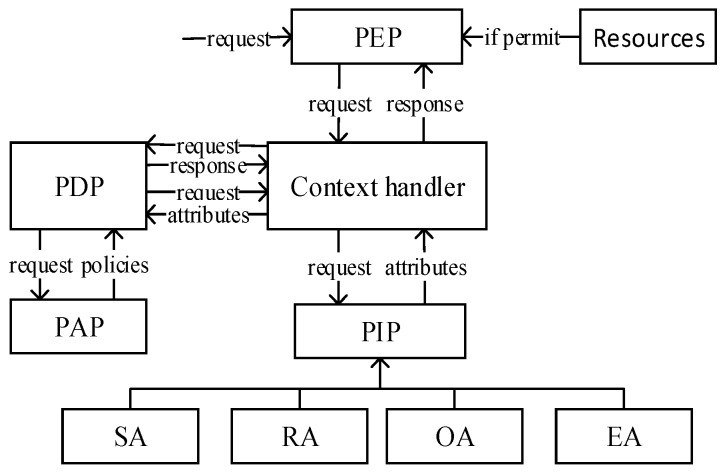
The data flow model of the XACML framework.

**Figure 2 sensors-20-01741-f002:**
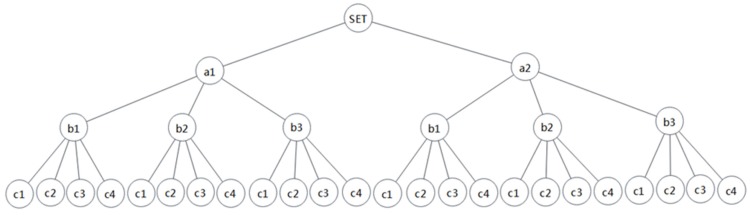
An example of a policy decision tree.

**Figure 3 sensors-20-01741-f003:**
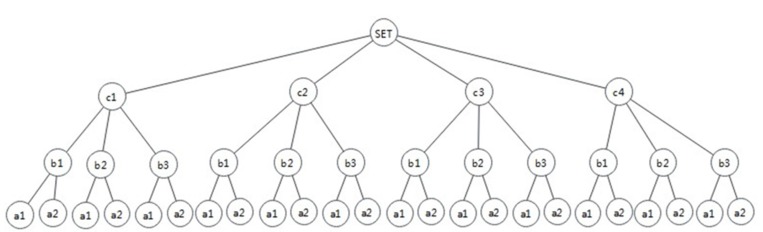
An example of the policy decision tree based on depth index.

**Figure 4 sensors-20-01741-f004:**
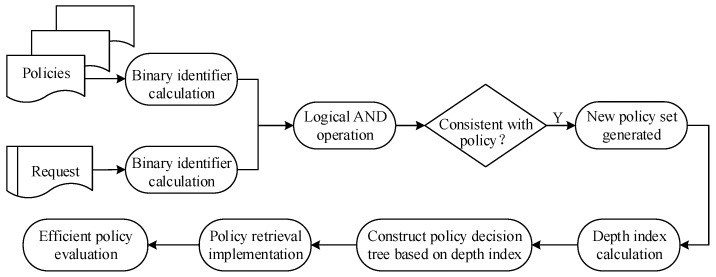
Policy evaluation process based on the attribute and value levels.

**Figure 5 sensors-20-01741-f005:**
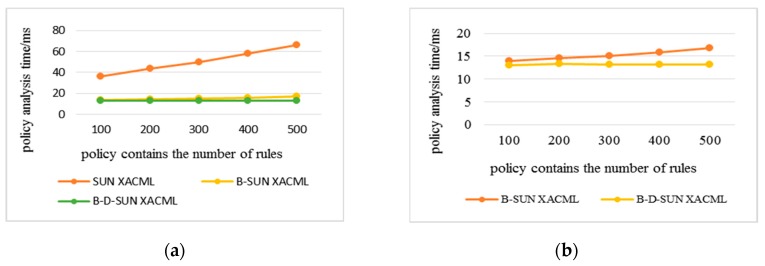
Policy analysis time varies with the number of rules. (**a**) Comparison of B-Sun XACML and B-D-Sun XACML with Sun XACML; (**b**) Detailed comparison between B-Sun XACML and B-D-Sun XACML.

**Figure 6 sensors-20-01741-f006:**
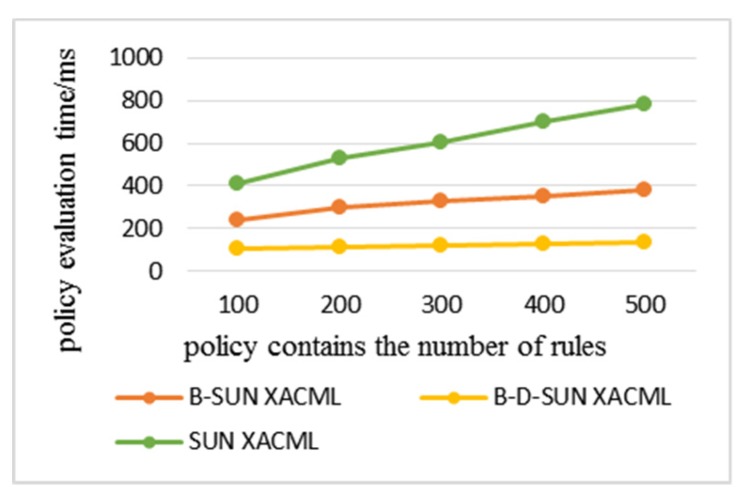
Policy evaluation time varies with the number of rules.

**Figure 7 sensors-20-01741-f007:**
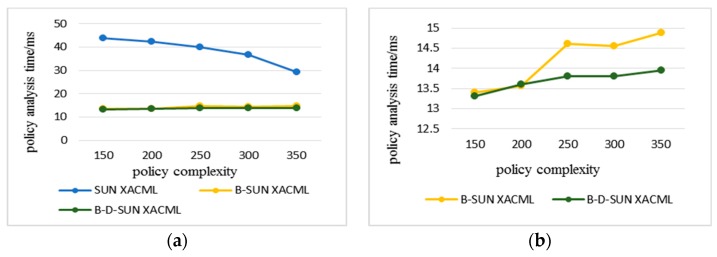
The policy analysis time varies with the increasing complexity of the policy. (a) Comparison of B-Sun XACML and B-D-Sun XACML with Sun XACML; (b) Detailed comparison between B-Sun XACML and B-D-Sun XACML.

**Figure 8 sensors-20-01741-f008:**
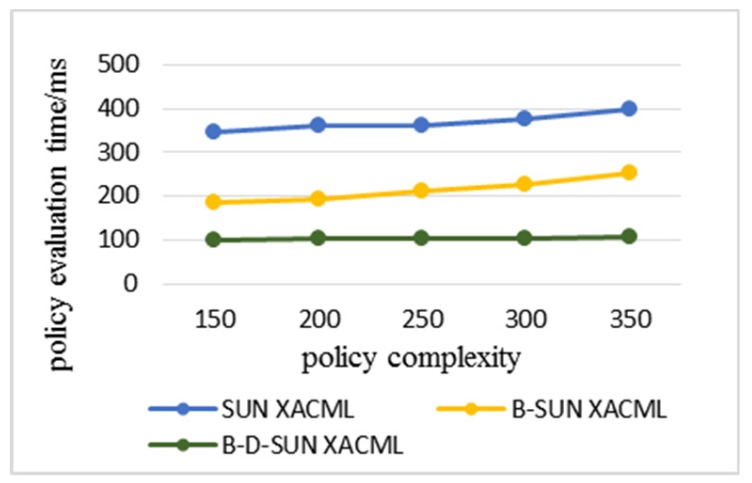
Policy evaluation time varies with the increasing complexity of the policy.

**Figure 9 sensors-20-01741-f009:**
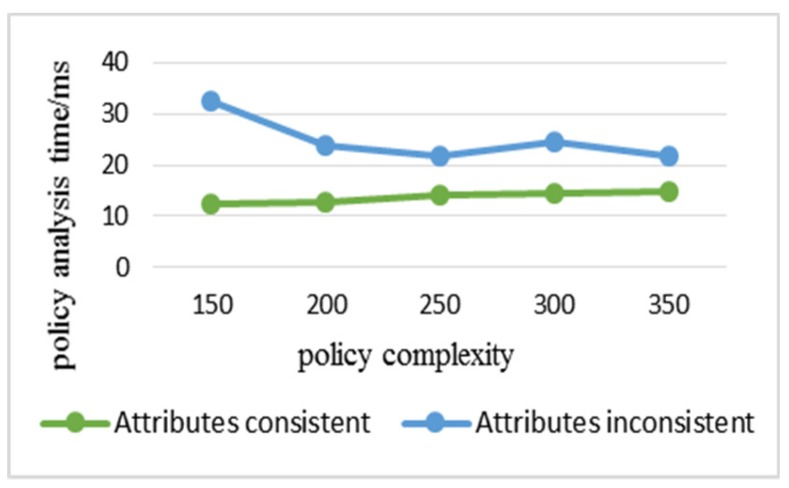
Time consumption in the case of attribute matching and mismatch.

**Table 1 sensors-20-01741-t001:** Set of attribute values for each attribute.

SA_ role = {Guest, common member, VIP}
SA_ level = {L0, L1, L2, L3, L4, L5}
SA_ terminal = {TV, Web, PC, Mobile}
RA_ type = {6-minute plot, standard plot, no ad, no titles, and tail}
RA_ category = {movie, TV series, variety, animation}
RA_ quality = {smooth, SD, HD, BD}
OA_ type = {watch, upload, download}
EA_ network= {local connection, broadband connection}

**Table 2 sensors-20-01741-t002:** Policy description and corresponding binary identifier.

Policy Description	Identifier
Guest can watch 6-minutes plot at any terminal other than the web	10111010
Common member can watch any standard plot in SD format on PC	10110110
L0 VIP users cannot download movies in BD format	11001110
VIP users can watch video without titles and tail of any quality	10010010

**Table 3 sensors-20-01741-t003:** Policy decision tree retrieval scheme.

Retrieval Order	Each Depth Retrieval Number	Total Retrieval Number
A–B–C	24 → 12 → 4	40
A–C–B	24 → 12 → 3	39
B–A–C	24 → 8 → 4	36
B–C–A	24 → 8 → 2	34
C–A–B	24 → 6 → 3	33
C–B–A	24 → 6 → 2	32

**Table 4 sensors-20-01741-t004:** Proportion of avp mismatch under different policy complexity.

Policy Complexity	Total Number of Mismatches	Average Number of Mismatches	Proportion
150	278	55.6	37%
200	318	63.6	31.80%
250	286	57.2	22.90%
300	266	53.2	17.70%
350	168	33.6	9.60%

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
