# Peer review of "An Efficient Attribute-Based Access Control (ABAC) Policy Retrieval Method Based on Attribute and Value Levels in Multimedia Networks"

_sensors, 2020, doi:10.3390/s20061741_

Round 1
Reviewer 1 Report
In this paper, the authors proposed an ABAC policy evaluation method based on the existing XACML framework. They used a policy decision tree retrieval method based on the depth index at the attribute value level. Comments for authors:- The title does not clarify if the focus of this paper is on access policy or operational policy.
- For the policy itself, it is not clear if the emphasis is on policy generation, analysis, retrieval, and/or evaluation.
- Then, what is the goal of the policy evaluation here: multimedia data security/privacy? access control? or multimedia communications?
- Authors used the term IoMT, however, did not clearly define it - what is smart heterogeneous multimedia things within the context of this paper? How is it related to various multimedia applications? Can you give examples of the context and content? Lack of clear details also makes me wonder if it is truly an IoT paper or a multimedia paper that has been revamped to give an IoT flavor.
- The literature work and the background review seem scattered and at times did not seem related to paper. The corresponding descriptions also lacked details to understand those prior works and hence the motivation and necessity of the paper.
- The manuscript did not provide sufficient details of the objects, algorithms and experimental setup to replicate the proposed approach by prospective readers. The simulation and the results are trivial.
- English must be improved throughout the manuscript. One example is: "Only when all the information matches, will the 176 corresponding authorities be given, so it is inefficient.”
- Overall, I think the organization of the paper played a role in my decision - contents and contexts are not coherent. Therefore, the use of subsections to compartmentalize the concepts introduced would have been helpful. The novel contribution and the necessity of the proposed work need more emphasis and clear details. I recommend the authors to provide more examples and descriptions of the background work based on which the authors built their work to make the paper self-contained. Last but not least, simulation-based results are not reliable in the given contexts for obvious reasons unless the code/setup has been made public for others to verify.
Author Response
Please see the attachment, thank you.

Reviewer 2 Report
The paper described a new method for performance increase on access control policies.
I consider the title not properly describing the research: "An Efficient Policy Evaluation Method Based on Attribute and Value Levels in Multimedia Network". It should be stated that there are access control policies. Also "evaluation" has different meaning, maybe "processing" is more suitable English language needs corrections (please read phrase by phrase, sometimes the sentences need to be restructured/ rephrased) The simulation environment and methodology should be better described More examples of multimedia policies (attributes, values) should be described, so the IoMT usecase is justified. Why is this performance improvement suitable for multimedia, compared to other domains? How often are these policies evaluated in a real multimedia case. The references should have a unitary style, some include the complete author first name, some start with family name ... some URLs are not properly cited.Author Response
Please see the attachment, thank you.

Round 2
Reviewer 2 Report
I appreciate the authors response in order to comply with previous recommandations. The paper in improved and also English language is corrected.